# Synergistic Cytokine Production by ATP and PGE_2_ via P2X4 and EP_3_ Receptors in Mouse Bone-Marrow-Derived Mast Cells

**DOI:** 10.3390/cells11040616

**Published:** 2022-02-10

**Authors:** Kosuke Obayashi, Kazuki Yoshida, Masa-aki Ito, Tetsuya Mori, Kimiko Yamamoto, Toshiyashu Imai, Isao Matsuoka

**Affiliations:** 1Laboratory of Pharmacology, Faculty of Pharmacy, Takasaki University of Health and Welfare, Takasaki 370-0033, Gunma, Japan; 0821024@takasaki-u.ac.jp (K.O.); yoshida-k@takasaki-u.ac.jp (K.Y.); mito@takasaki-u.ac.jp (M.-a.I.); 2Laboratory of Allergy and Immunology, Faculty of Pharmacy, Takasaki University of Health and Welfare, Takasaki 370-0033, Gunma, Japan; tmori@takasaki-u.ac.jp; 3Department of Biomedical Engineering, Graduate School of Medicine, The University of Tokyo, Tokyo 113-0033, Japan; kyamamoto@m.u-tokyo.ac.jp; 4Discovery Research Laboratories, Nippon Chemiphar Co., Ltd., Misato 341-0005, Saitama, Japan; t-imai@chemiphar.co.jp

**Keywords:** ATP, bone-marrow-derived mast cells, prostaglandin E_2_, P2X4 receptor, EP_3_ receptor

## Abstract

ATP is an important intercellular messenger in the extracellular space. In mast cells (MCs), ATP stimulates the ionotropic P2X4 receptor (P2X4R), resulting in enhanced degranulation and exacerbation of acute allergic reactions. In this study, we investigate whether ATP regulates inflammatory cytokine production in MCs. Gene expression was analyzed by quantitative RT-PCR, and cytokine production was measured using ELISA. The stimulation of mouse bone-marrow-derived MCs (BMMCs) with ATP alone had little effect on cytokine secretion. However, the co-stimulation with prostaglandin (PG) E_2_ resulted in a marked increase in the secretion of various cytokines, such as tumor necrosis factor-α, interleukin (IL)-6, and IL-13, accompanied by an increase in their mRNA levels. The effects of ATP were inhibited by P2X4R antagonists and diminished in BMMCs derived from P2X4R-deficient mice, suggesting that P2X4R mediated the reaction. The effects of PGE_2_ were mimicked by an EP_3_ receptor (EP_3_R) agonist and blocked by an EP_3_R antagonist. The synergistic cytokine mRNA elevations induced by ATP and PGE_2_ were blocked by nuclear factor-κB and Ca^2+^-calcineurin signaling inhibitors. Altogether, these results suggest that combining P2X4R and EP_3_R signaling enhances acute degranulation and the subsequent cytokine secretion, exacerbating allergic inflammation.

## 1. Introduction

Adenosine triphosphate (ATP), an important intracellular energy source constantly supplied by cell metabolism under cellular respiration, is present in high concentrations in the cytosol. However, cells release ATP into the extracellular space as an intercellular mediator to regulate various physiological functions [1]. Extracellular ATP accumulation is recognized by various receptors called P2 receptors. Seven subtypes of ionotropic P2X receptors (P2X1-7) recognize ATP exclusively and open non-selective cation channels [2]. In contrast, eight subtypes of G protein-conjugated P2Y receptors have been identified in humans, including P2Y_1_, Y_2_, Y_4_, Y_6_, and Y_11–14_ receptors, and the ligands are not only ATP but also ADP and uridine nucleotides, such as UTP, UDP, and UDP-glucose [3]. Receptor activation is transmitted into cells by the G protein-dependent signaling [3]. In addition, extracellular ATP is rapidly dephosphorylated by various extracellular ATP-degrading enzymes to adenosine, acting as an agonist to another set of P1 adenosine receptors, such as Gi-coupled A_1_ and A_3_ receptors, and Gs-coupled A_2A_ and A_2B_ receptors [4]. Thus, ATP is involved in regulating physiological functions as a spatially and temporally diverse intercellular transmitter.

Initially, ATP was focused as a transmitter of non-cholinergic and non-adrenergic neurotransmission in the regulation of the contraction of smooth muscle organs, such as the intestine [5], bladder [6], and seminal vesicle [7]. However, tissue damage or mechanical stimulation causes a large amount of ATP release as a danger or find-me signal [8], facilitating the accumulation of evidence that ATP plays an important role in regulating immune cells. For example, ATP and UTP released from apoptotic cells stimulate P2Y and adenosine receptors on neutrophils [9], macrophages [10], and lymphocytes [11] to promote migration to the affected area. In macrophages, the stimulation of P2X7 receptors activates the inflammasome, promoting IL-1β processing and inducing IL-1β release [12]. Activating P2X4 receptors (P2X4R) in microglia plays an essential role in inducing neuropathic pain [13].

Mast cells (MCs) express various P1 and P2 purinergic receptors, and important information about the role of purinergic signaling in MC function has been reported [14]. For example, the stimulation of MCs with high concentrations of ATP induces direct degranulation via P2X7 receptors involved in the pathophysiology of inflammatory bowel disease [15] and skin hypersensitivity [16]. In addition, adenosine promotes antigen-stimulated degranulation via the A_3_ receptor [17]. G protein-coupled P2Y receptors are also abundantly expressed in MCs; however, their effects have been reported only for promoting antigen-induced degranulation via the stimulation of P2Y_13_ and P2Y_14_ receptors in model cell lines [18,19]. Recently, we reported that ATP promotes antigen-stimulated degranulation via ionotropic P2X4R using mouse bone-marrow-derived MCs (BMMCs) [20]. Furthermore, the P2X4R stimulating effect increased the antigen-independent degranulation reaction [21]. Co-stimulation with the EP_3_ receptor (EP_3_R) of prostaglandin (PG) E_2_ coupled to the Gi protein and the A_3_ receptor of adenosine induces a remarkable degranulation reaction. Such enhancement of antigen-dependent and antigen-independent degranulation reactions affected the degree of allergic reaction in vivo [20]. Since the magnitude of the degranulation reaction is related to the severity of the immediate allergic reaction, an increase in the P2X4R-mediated degranulation reaction is considered important for understanding allergic reactions.

MCs induce an immediate allergic reaction by releasing granule contents, such as histamine, serotonin, and various proteases, and produce various chemokines and cytokines as a delayed reaction [22]. Compared to the regulatory mechanism of the degranulation reaction, the purinergic regulation of cytokine production in MCs remains unexamined. In this study, we investigate whether extracellular ATP affects the production of cytokines by antigen and non-antigen stimulation using BMMCs.

## 2. Materials and Methods

### 2.1. Materials

ATP, α,β-methylene ATP (αβmeATP), ADP, UTP, PGE_2_, dexamethasone, 2,4-dinitrophenyl human serum albumin (DNP-HSA), anti-DNP IgE (clone SPE-7), cyclosporine and the GenElute Mammalian Total RNA miniprep kit were obtained from Sigma-Aldrich (Tokyo, Japan). 1,2-Bis (2-aminophenoxy)ethane-N,N,N′,N′-tetraacetic acid tetrakis acetoxymethyl ester (BAPTA-AM) was obtained from Abcam (Cambridge, UK). Allophycocyanin-conjugated rat anti-mouse c-Kit antibodies (clone 2B8) were obtained from BD Pharmingen (Tokyo, Japan). Phycoerythrin-conjugated mouse anti-mouse FcεRIα antibodies (clone MAR-1) were obtained from eBioscience (San Diego, CA, USA). Recombinant mouse interleukin (IL)-3 and recombinant mouse SCF were obtained from Peprotech (London, UK). SB203580, wortmannin and AH6809 were obtained from Cayman Chemical (Ann Arbor, MI, USA). InSolution NF-kB activation inhibitor was obtained from Calbiochem (San Diego, CA, USA). U0126, anti-extracellular signal-regulated kinase (ERK)1/2, anti-phospho-ERK1/2, anti-Akt, anti-phospho-Akt (Thr308), anti-NF-κB p65 and anti-phospho-NF-κB p65 (Ser536) antibodies were obtained from Cell Signaling Technology (Danvers, MA, USA). Anti-P2X4 antibody was obtained from Alomone Labs (Jerusalem, Israel). Mouse IL-1β, IL-6, IL-13, and tumor necrosis factor (TNF)-α enzyme-linked immunosorbent assay (ELISA) kits were obtained from Thermo Fisher Scientific (Tokyo, Japan). ONO-DI-004 (EP_1_ agonist), ONO-AE1-259-01 (EP_2_ agonist), ONO-AE-248 (EP_3_ agonist), ONO-AE1-329 (EP_4_ agonist), ONO-8713 (EP_1_ antagonist), ONO-AE3-208 (EP_3_ antagonist), and ONO-AE3-240 (EP_4_ antagonist) were obtained from ONO Pharmaceuticals Co., Ltd. (Osaka, Japan). NP-1815-PX (P2X4 antagonist) was provided by Nippon Chemiphar Co., Ltd. (Tokyo, Japan). All other chemicals were of reagent grade or the highest quality available.

### 2.2. Animals

P2X4R-deficient (*P2rx4^−^*^/−^) mice were generated by Dr. Yamamoto (University of Tokyo), as described previously [23]. C57BL/6 mice were purchased from SLC Japan (Hamamatsu, Japan). All mice were maintained in the animal facility of the Takasaki University of Health and Welfare under specific pathogen-free conditions. All experiments were performed per the Animal Research Committee of Takasaki University of Health and Welfare regulations (Approval No. 2033, 1 April 2020).

### 2.3. Cell Culture

BMMCs were prepared from the bone marrow of C57BL/6 mice, as described previously [24]. Briefly, bone marrow cells were collected from the femur and cultured in RPMI 1640 growth medium containing 10% fetal bovine serum, 100 units/mL penicillin, 100 μg/mL streptomycin, and 10 ng/mL mouse IL-3. After 2 weeks, recombinant SCF (10 ng/mL) was added to the growth medium and cultured for 2–3 weeks. The culture medium was changed twice per week. After these treatments, almost all (> 95%) cells displayed an MC phenotype, as indicated by CD117 (c-Kit), and FcεRI expression measured using a FACSCant II flow cytometer (BD Biosciences, Tokyo, Japan).

### 2.4. Measurement of Cytokine Secretion

BMMCs were sensitized with 50 ng/mL anti-DNP IgE overnight in RPMI 1640 growth medium. Cells were washed twice with RPMI medium containing 0.1% bovine serum albumin and then stimulated under various conditions for different periods at 37 °C. The reactions were terminated by centrifugation, and the supernatant and cell pellets were subjected to cytokine measurement using ELISA and RNA extraction, respectively.

### 2.5. Quantitative RT-PCR

Total RNA was isolated using the GenElute Mammalian Total RNA miniprep kit. First-strand cDNA was synthesized using Moloney murine leukemia virus reverse transcriptase with a 6-mer random primer. Quantitative RT-PCR was performed using an SYBR green kit (Takara bio, Tokyo Japan), as described previously [25].

### 2.6. Western Blotting

Cells were collected, washed twice with Krebs-Ringer-HEPES buffer (130 mM NaCl, 4.7 mM KCl, 4.0 mM NaHCO_3_, 1.2 mM KH_2_PO_4_, 1.2 mM MgSO_4_, 1.8 mM CaCl_2_, and 11.5 mM glucose and 10 mM HEPES (pH 7.4)), and then stimulated under various conditions at 37 °C. The reactions were terminated by adding 4x Laemmli sample buffer. The lysate was separated by 10 % sodium dodecyl sulfate-polyacrylamide gel electrophoresis (SDS-PAGE) and transferred to Immobilon-P polyvinylidene fluoride membranes. The membranes were blocked with 5% non-fat milk for 1 h at room temperature and exposed to primary antibodies overnight at 4 °C and then to secondary antibodies for 2 h at room temperature. The antibodies were diluted as follows: anti-phospho-ERK1/2 (1:1000), anti-ERK1/2 (1:1000), anti-phospho-Akt (1:1000), anti-Akt (1:1000), anti-phospho-NF-κB p65 (1:1000), anti-NF-kB p65 (1:1000), anti-P2X4 (1:2000), and horseradish peroxidase (HRP)-linked anti-rabbit IgG (1:10,000). Immunoreactive proteins were detected by enhanced chemiluminescence (GE Healthcare Bio-sciences, Tokyo, Japan) using Image Reader LAS-3000 (FUJIFILM, Tokyo, Japan). The density of bands was quantified using Multi Gauge version 3.0, and the fold change in phosphorylation was calculated from the amount of phosphoprotein relative to total proteins as a loading index.

### 2.7. Statistics

All experiments were repeated at least three times, yielding similar results. Data represent the mean ± standard error of the mean (SEM). Statistical analyses were performed using Student’s *t*-test for two-sample comparisons and one-way analysis of variance (ANOVA) with Dunnett’s two-tailed test for multiple comparisons. Statistical significance was set at *p*-value < 0.05.

## 3. Results

### 3.1. Effects of ATP Combined with PGE_2_ and Ag on Cytokine mRNA Expression and Secretion in BMMCs

First, we examined the effects of ATP on PGE_2_ or antigen DNP-HSA-induced mRNA expression of inflammatory cytokines, including IL-1β, IL-6, IL-13, and TNF-α, in BMMCs by quantitative RT-PCR. The BMMCs used in this study hardly expressed the inflammatory cytokine mRNA in the unstimulated condition. However, the co-stimulation of ATP (100 μM) with PGE_2_ (1 μM) and low doses of antigen DNP-HSA (10 ng/mL) markedly increased cytokine mRNA expression (Figure 1). The increased levels were comparable to the expression levels of glyceraldehyde-3-phosphate dehydrogenase (GAPDH), a housekeeping gene used as an internal standard. Next, we examined whether the change in mRNA reflected cytokine secretion. The stimulation of the BMMCs for 3 h with ATP alone had little effect on the production of IL-6, IL-13, and TNF-α; however, co-stimulation with antigen or PGE_2_ resulted in a pronounced increase in cytokine secretion. Both the rate and absolute value of the increase were more pronounced when co-stimulated with PGE_2_ than with antigen (Figure 2). In contrast, there was almost no IL-1β secretion, exhibiting increased mRNA levels (data not shown). Therefore, the subsequent experiments dealt with a detailed analysis of the production of IL-6, IL-13, and TNF-α by co-stimulation with ATP and PGE_2_.

### 3.2. Time-Dependent Increase in IL-6 and TNF-α Secretion in Response to ATP, PGE_2_, and Their Co-Stimulation

Although an increase in cytokine secretion is accompanied by de novo synthesis in response to stimuli, some cytokines, such as TNF-α, are exited as granule content and increased immediately after stimulation. Therefore, the detailed time-course of changes in cytokine secretion was investigated (Figure 3). ATP stimulation alone did not induce significant cytokine secretion until 24 h. In contrast, PGE_2_ stimulated IL-13 and TNFα secretion at 3 h after stimulation and IL-6 secretion at 6 h. When co-stimulated with ATP and PGE_2_, a marked increase in IL-6 and IL-13 secretion was observed 3 h after a 1 h time lag. In contrast, TNFα secretion was increased 1 h after co-stimulation with ATP and PGE_2_ and reached a peak secretion at 3 h. Based on the above results, the following cytokine secretion was examined 3 h after stimulation, whereas mRNA expression was determined 1 h after stimulation.

### 3.3. Effects of Different Concentrations of ATP and PGE_2_ on IL-6, IL-13, and TNF-α Secretion

The effects of different concentrations of ATP and PGE_2_ on IL-6, IL-13, and TNF-α were examined. ATP alone did not release cytokines up to a concentration of 100 μM; however, the secretion of IL-6, IL-13, and TNF-α increased slightly but constantly at 1 mM. In the presence of PGE_2_ (1 μM), ATP increased the secretion of IL-6, IL-13, and TNFα in a concentration-dependent manner from 1 to 100 μM, and the enhancement was decreased at 1 mM. In contrast, PGE_2_ elicited IL-6, IL-13, and TNFα secretion at 0.1 to 1 μM even in the absence of ATP; however, its effects were markedly increased in the presence of 100 μM ATP (Figure 4). Based on these results, 100 μM ATP and 0.1 μM PGE_2_ were selected as optimal concentrations for assessing the synergistic secretion of cytokines in the subsequent experiments.

### 3.4. Effects of Different Purinergic Receptor Agonists Combined with PGE_2_ on IL-6, IL-13, and TNF-α mRNA Expression

The BMMCs used in our studies expressed multiple P2 receptors stimulated by ATP, including P2X1, P2X4, P2X7, P2Y_1_, P2Y_2_, and P2Y_13_ [24]. To examine the receptor subtype involved in ATP enhancement of PGE_2_-induced cytokine secretion, the effects of various nucleotide agonists were examined. The agonists used were P2X1 agonist α,β-methylene ATP (αβmeATP), P2Y_1_ and P2Y_13_ agonist ADP, and P2Y_2_ agonist UTP. Among them, ADP enhanced the secretion of IL-6 and IL-13, but was much weaker than ATP. αβmeATP and UTP did not increase the effects of PGE_2_ (Figure 5).

### 3.5. Role of P2X4 Receptor in Enhancing Cytokine mRNA Expression and Secretion by Co-Stimulation of ATP and PGE_2_

The responses to P2 agonists shown in Figure 5 were similar to the effects on antigen- and PGE_2_-induced MC degranulation via P2X4R. Therefore, the effect of NP-1815-PX, a P2X4R antagonist [26], on cytokine secretion was investigated. NP-1815-PX did not affect cytokine secretion by PGE_2_, but significantly suppressed the augmented response induced by ATP (Figure 6). To further clarify the involvement of P2X4R, we examined the BMMCs prepared from *P2rx4*^−/−^ mice. No increase in cytokine mRNA expression and cytokine secretion was observed in *P2rx4*^−/−^ BMMCs after co-stimulation with ATP and PGE_2_ (Figure 7).

### 3.6. Role of EP_3_R in Enhancing Cytokine mRNA Expression and Secretion by Co-Stimulation of ATP and PGE_2_

We examined the receptors that mediate PGE_2_ action. The synergistic increase in cytokine secretion by the co-stimulation of PGE_2_ and ATP was reproduced only with the EP_3_R agonist ONO-AE-248, and EP_1_, EP_2_, and EP_4_ receptor agonists showed no effect (Figure 8A–C). Conversely, the synergistic increase in cytokine secretion by the co-stimulation of PGE_2_ and ATP was suppressed by the EP_3_R antagonist ONO-AE3-240, but not by EP_1_, EP_2_, and EP_4_ receptor antagonists (Figure 8D–F). Consistently, the BMMCs predominantly expressed EP_3_R mRNA, which was unaffected by P2X4R deficiency (Appendix A).

### 3.7. Effects of ATP and PGE_2_ on ERK/Akt/NF-κB Signaling Pathway

The stimulation of the BMMCs with PGE_2_ induced _the_ rapid phosphorylation of both ERK1/2 and Akt within 5 min, followed by a decrease to a sustained level at 10 and 20 min. In contrast, ATP had little effect on ERK1/2 and Akt phosphorylation at 5 min, but slightly increased after 10 and 20 min. The co-stimulation of ATP and PGE_2_ induced a rapid and sustained increase in ERK1/2 and Akt phosphorylation at 10 and 20 min (Figure 9A,C,D). These co-stimulating effects of ATP and PGE_2_ on ERK1/2 and Akt phosphorylation were absent in *P2rx4*^−/−^ BMMCs (Figure 9B–D). Akt is known to activate nuclear factor-κB (NF-κB) [27], a transcription factor for various inflammatory cytokines. ATP or PGE_2_ alone only slightly increased the level of phosphorylated NF-κB p65, an activated form of NF-κB. The co-stimulation with ATP and PGE_2_ markedly increased the levels of phosphorylated NF-κB (Figure 9A,E). The synergistic phosphorylation of NF-κB by ATP and PGE_2_ was absent in *P2rx4*^−/−^ BMMCs (Figure 9B,E). In another set of analysis, we examined the role of P2X4R in the synergistic increase in the phosphorylation of ERK1/2, Akt, and NF-kB by co-stimulation with ATP and PGE_2_. P2X4R deficiency abolished the increased phosphorylation of ERK, Akt, and NF-κB (Appendix A).

### 3.8. Effects of Signal Transduction Pathway Inhibitors on the Enhanced Cytokine mRNA Expression by Co-Stimulation with ATP and PGE_2_

The effects of various inhibitors against ERK, p38MAPK, phosphatidylinositol 3-kinase (PI3K), calcineurin, and NF-κB were examined. The elevation of cytokine mRNA levels by the co-stimulation of ATP and PGE_2_ was attenuated by the ERK inhibitor U0126, PI3K inhibitor wortmannin, NF-κB activation inhibitor NFκBI, calcineurin inhibitor cyclosporine A, intracellular Ca^2+^ chelator BAPTA-AM and the steroidal anti-inflammatory drug dexamethasone. In contrast, the p38MAPK inhibitor SB203580 did not suppress cytokine mRNA expression (Figure 10)

## 4. Discussion

In a previous study, we reported that ATP augments IgE-dependent and IgE-independent MC degranulation via P2X4R activation [20,21]. MCs elicit an immediate allergic reaction by releasing chemical mediators stored in the granules; however, in the delayed phase, MCs subsequently secrete diverse inflammatory cytokines and chemokines, causing the migration and activation of other immune cells, resulting in an allergic inflammation [22]. The present study showed that extracellular ATP markedly increased the gene expression and secretion of inflammatory cytokines IL-6 and TNF-α and Th-2 cytokine IL-13 by co-stimulation with antigen and PGE_2_. Pharmacological analysis using agonists and antagonists revealed that the effects of ATP and PGE_2_ were mediated by P2X4R and EP_3_R, respectively, as similar to the effects on degranulation responses [20,21]. The involvement of P2X4R in the ATP-induced effect was confirmed using BMMCs prepared from *P2r**x**4^−/−^* mice. Regarding enhancing the degranulation reaction by ATP, the maximum response was not significantly different between the co-stimulation with antigen and with PGE_2_ [20,21]. However, the maximum cytokine secretion was much larger in the co-stimulation of ATP with PGE_2_ than that with the antigen. FcεRI activation with antigen is triggered by tyrosine phosphorylation of the immunoreceptor tyrosine-based activation motif (ITAM) in the FcεRI β subunit. It recruits and activates Lyn kinase resulting in Syk kinase activation, propagating phosphorylation-dependent signals to activate various effector proteins, including phospholipase Cγ [28]. Although FcεRI β subunit is indispensable for FcεRI signaling, the ITAM in the β subunit has an inhibitory regulatory role in MC cytokine production [29], explaining the low cytokine secretion combined with ATP and antigen. In this study, we focused on increased cytokine secretion by co-stimulation with PGE_2_ and ATP.

Although various ATP receptors are expressed in MCs, the receptor that mediates the ATP-induced enhancement of cytokine secretion in the presence of PGE_2_ was suggested to be the P2X4 receptor, as described above. No increase in cytokine secretion with ATP in the BMMCs prepared from *P2rx4*^−/−^ mice and the failure of αβmeATP and UTP to stimulate cytokine secretion indicates that P2X1 and P2Y_2_ receptors expressed abundantly in the BMMCs [24] do not affect cytokine secretion. Among the P2 receptor agonists tested, ADP slightly but significantly enhanced the gene expression of IL-6 and IL-13. Unlike ATP, ADP did not enhance TNF-α expression; hence, it was not the result of P2X4R stimulation with low affinity but of stimulating the Gq-coupled P2Y_1_ receptor that recognizes ADP. This result cannot infer that Gq-mediated signals enhance cytokine production, and it is necessary to activate other G proteins, such as G_12/13_, in addition to Gq [30]. Further studies are needed to validate this hypothesis.

The stimulation of the P2X4 receptor, an ion channel-type receptor, activates the non-selective cation channel, leading to Ca^2 +^ influx and depolarization of the membrane potential [31]. It is interesting to understand such a stimulating mechanism affecting the gene expression of inflammatory cytokines. The effect on cytokine production was markedly different among the P2X receptor subtypes expressed in BMMCs. For example, P2X1 receptor stimulation with αβmeATP has no effect, whereas P2X7 receptor stimulation with a high ATP concentration (~1 mM) alone slightly, but significantly stimulated IL-6, IL-13, and TNF-α secretion. However, unlike the P2X4R-mediated effect, activating the P2X7 receptor did not enhance PGE_2_-induced cytokine secretion, but rather inhibited P2X4R-mediated effects. In addition to ion channel function, P2X7 receptor stimulation dilate the channel pore, allowing it to pass with high molecular weight substances, affecting the MC membrane function [32]. The three ionotropic P2X1, P2X4, and P2X7 receptors are functionally expressed in human MCs [33]. The difference in the effects of these P2X receptors on cytokine secretion may be useful in elucidating the unique mechanism of P2X4R action.

Increased cytokine secretion by P2X4R stimulation combined with PGE_2_ was accompanied by increased mRNA levels. The time course of cytokine secretion showed an increase in gene expression before the increase in the secretion of all three cytokines examined. Cytokine secretion was not remarkable 1 h after stimulation when a marked mRNA increase was detected. IL-6, IL-13, and TNF-α secretion increased rapidly during 1–3 h after stimulation, and no continuous increase was observed thereafter, suggesting that P2X4R signaling enhances the transcription of inflammatory cytokines. NF-κB is involved in the transcription of various inflammatory cytokines [22,34]. NF-κB is present in the cytosol as a complex with IκB. When IκB is phosphorylated by IKK, ubiquitination of IκB progresses its degradation by the proteasome, and the dissociated NF-κB translocate into the nucleus to stimulate inflammatory cytokine gene expression [35,36]. Among EP_3_R downstream signals, ERK1/2 and PI3K-Akt activation participates in NF-κB signaling pathway [26]. PGE_2_ rapidly increased ERK1/2 and Akt phosphorylation, followed by a decrease to low steady levels. In addition, ATP increased phosphorylation of ERK1/2 and Akt in a time-dependent manner, but this action remained in *P2rx4*^−/−^ BMMCs, suggesting effects other than P2X4R. However, our results showed that ATP did not affect PGE_2_-induced ERK1/2 and Akt phosphorylation in the early period (5 min), but significantly increased them at later periods (10–20 min). Importantly, this enhancement of PGE_2_-induced ERK1/2 and Akt phosphorylation by ATP was absent in *P2rx4*^−/−^ BMMCs. Furthermore, the PI3K inhibitor WMN, the ERK inhibitor U0126, and the NF-κB inhibitor attenuated the upregulation of inflammatory cytokine expression by ATP in the presence of PGE_2_. These results suggest that P2X4R stimulation enhances the ERK1/2 and PI3K/Akt/NF-κB signaling pathway and increases inflammatory cytokine gene expression. Consistent with this hypothesis, the steroidal anti-inflammatory dexamethasone, which potently inhibits the NF-κB pathway [37], also inhibited inflammatory cytokine mRNA expression induced by ATP and PGE_2_. More direct evidence was observed with the change in phosphorylated NF-kB p65, a form of activated NF-κB. Namely, ATP or PGE_2_ alone induced only a slight increase in phosphorylated NF-κB p65, but their combination markedly increased the phosphorylated NF-κB p65. In addition, this effect was absent in *P2rx4^−/−^* BMMCs. The phosphorylation of NF-κB p65 by ATP and PGE_2_ was rapidly induced within 5 min and returned to basal levels in 20 min. These changes are consistent with the rapid induction of cytokine mRNAs. This also suggests that sustained elevations of ERK and Akt after 10 min are not responsible for NF-κB phosphorylation but may support NF-κB-dependent cytokine gene transcription.

In addition to NF-κB-dependent signaling, a recent study reported by Jordan et al. [38] demonstrated that Ca^2+^-calcineurin signaling is involved in cytokine secretion induced by co-stimulation with ATP and IL-33. In this study, we also showed that the intracellular Ca^2+^ chelating agent BAPTA-AM and the calcineurin inhibitor cyclosporine A inhibited inflammatory cytokine mRNA expression induced by ATP and PGE_2_. P2X4R-mediated Ca^2+^ influx may stimulate calcineurin-dependent NFAT signaling, resulting in increased inflammatory cytokine gene transcription. Since P2X1 and P2X7 receptors, which input signals qualitatively similar to P2X4R, did not show interplay with EP_3_R signaling, P2X4R may transmit unspecified and unique signals, which is considered an important issue that needs to be addressed.

## 5. Conclusions

The present study revealed that extracellular ATP enhanced inflammatory cytokine secretion from stimulated MCs via P2X4R activation. This enhancement was more pronounced with co-stimulation with PGE_2_ than with antigen. Since both PGE_2_ and ATP are commonly accumulated in the inflamed microenvironment, these mediators should affect MC function in various physiological and pathophysiological situations. The cross-talk between P2X4R and EP_3_R signals upregulates an acute degranulation response and following inflammatory cytokine secretion, suggesting that these receptors may be potential therapeutic targets for diseases caused by MCs.

## Figures and Tables

**Figure 1 cells-11-00616-f001:**
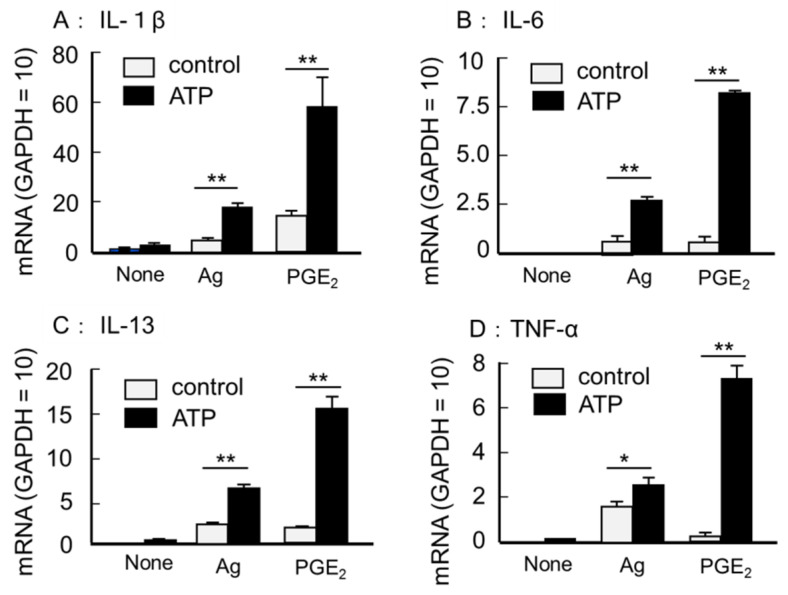
Effects of ATP on antigen (Ag)- and PGE_2_-induced inflammatory cytokine mRNA expressed in the BMMCs. The BMMCs were stimulated with a vehicle (None), DNP-HSA (Ag, 10 ng/mL), or PGE_2_ (1 μM) in the presence or absence of ATP (100 μM) for 1 h. mRNA expression for (**A**) IL-1β, (**B**) IL-6, (**C**) IL-13, and (**D**) TNF-α were examined by quantitative RT-PCR. Data were normalized to GAPDH mRNA levels. Values are shown as mean ± SEM (*n* = 3). * *p* < 0.05, ** *p* < 0.01.

**Figure 2 cells-11-00616-f002:**
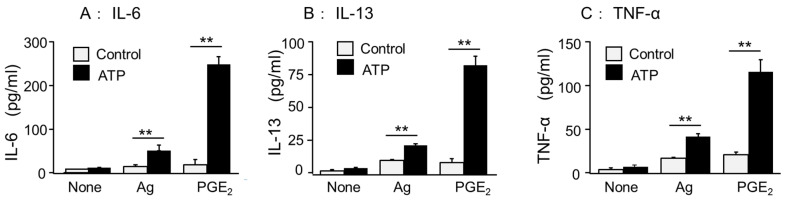
Effects of ATP on Ag- and PGE_2_-induced inflammatory cytokine secretion in the BMMCs. The BMMCs were stimulated with a vehicle (None), DNP-HSA (Ag, 10 ng/mL), or PGE_2_ (1 μM) in the presence or absence of ATP (100 μM) for 3 h. Cytokine levels of (**A**) IL-6, (**B**) IL-13, and (**C**) TNF-α were examined by ELISA. Values are shown as mean ± SEM (*n* = 3). ** *p* < 0.01.

**Figure 3 cells-11-00616-f003:**
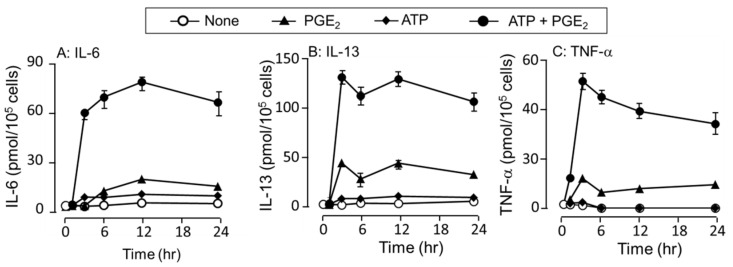
Effects of ATP on the time course of PGE_2_-induced cytokine secretion in the BMMCs. The BMMCs were stimulated with a vehicle (None), DNP-HSA (10 ng/mL) or PGE_2_ (1 μM) in the presence or absence of ATP (100 μM) for 1, 3, 6, 12, and 24 h. Cytokine levels of (**A**) IL-6, (**B**) IL-13, and (**C**) TNF-α were measured by ELISA. Values are shown as mean ± SEM from triplicate determination.

**Figure 4 cells-11-00616-f004:**
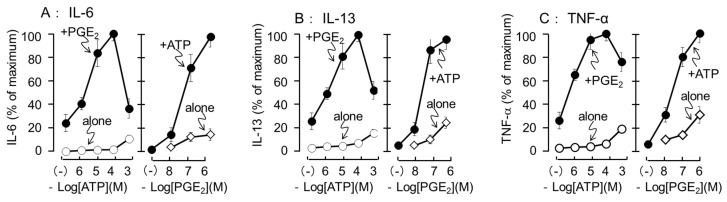
Effects of different concentrations of ATP or PGE_2_ on cytokine secretion induced by co-stimulation of ATP and PGE_2_ in the BMMCs. The BMMCs were stimulated with different concentrations of ATP (left) or PGE_2_ (right) in the presence or absence (alone) of PGE_2_ (1 μM) or ATP (100 μM), respectively, for 3 h. (**A**) IL-6, (**B**) IL-13, and (**C**) TNF-α levels in the reaction medium were measured by ELISA. Data are shown as percentage of maximum response obtained with ATP 100 μM + PGE_2_ (1 μM). Values are shown as mean ± SEM (*n* = 3).

**Figure 5 cells-11-00616-f005:**
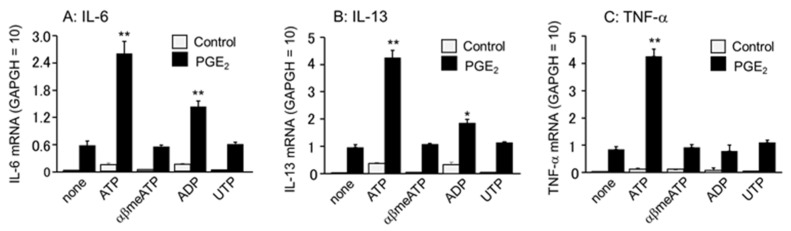
Effects of different P2 receptor agonists on PGE_2_-induced cytokine secretion in the BMMCs. The BMMCs were stimulated with a vehicle (None), ATP (100 μM), α,β-methylene ATP (αβmeATP, 10 μM), ADP (100 μM), or UTP (100 μM) in the presence (black column) or absence (open column) of PGE_2_ (0.1 μM) for 1 h. mRNA expression for (**A**) IL-6, (**B**) IL-13, and (**C**) TNF-α were examined by quantitative RT-PCR. Data were normalized with GAPDH mRNA levels. Values are shown as mean ± SEM (*n* = 3). * *p* < 0.05, ** *p* < 0.01 vs. PGE_2_ alone (none).

**Figure 6 cells-11-00616-f006:**
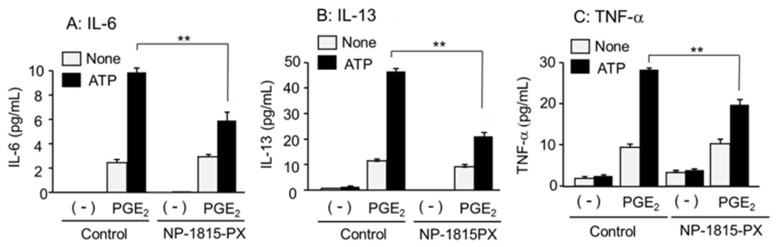
Effects of P2X4R antagonist NP-1815-PX on cytokine secretion induced by co-stimulation of ATP and PGE_2_ in the BMMCs. The BMMCs were preincubated vehicle (control) and NP-1815-PX (10 μM) for 10 min and then stimulated with vehicle (-) or PGE_2_ (0.1 μM) in the presence or absence of ATP (100 μM) for 3 h. (**A**) IL-6, (**B**) IL-13, and (**C**) TNF-α levels in the reaction medium were measured by ELISA. Values are shown as mean ± SEM (*n* = 3). ** *p* < 0.01.

**Figure 7 cells-11-00616-f007:**
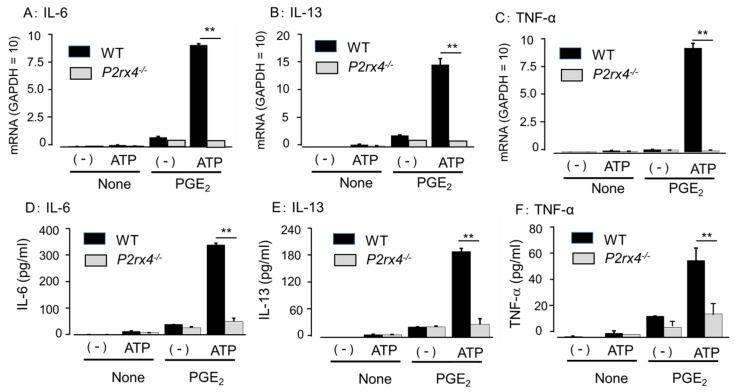
Lack of ATP-induced upregulation of PGE_2_-induced cytokine production in P2X4R-deficient (*P2rx4*^−/−^) mice. The BMMCs, prepared from wild-type (WT) and *P2rx4*^−/−^ mice, were stimulated with vehicle (None) or PGE_2_ (0.1 μM) in the presence or absence of ATP (100 μM) for (**A**–**C**) 1 h or (**D**–**F**) 3 h. mRNA expression for (**A**) IL-6, (**B**) IL-13, and (**C**) TNF-α were examined by quantitative RT-PCR. Data were normalized with GAPDH mRNA levels. Values are shown as mean ± SEM (*n* = 3). ** *p* < 0.01. The levels of (**D**) IL-6, (**E**) IL-13, and (**F**) TNF-α in the reaction medium were measured by ELISA. Values are shown as mean ± SEM (*n* = 3). ** *p* < 0.01.

**Figure 8 cells-11-00616-f008:**
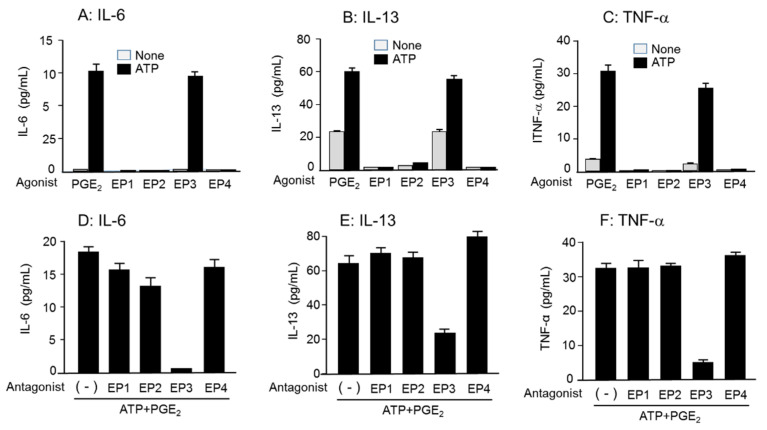
Involvement of EP_3_R in PGE_2_-induced enhanced cytokine secretion in the presence of ATP in BMMCs. (**A**–**C**) The BMMCs were stimulated with PGE_2_, ONO-DI-004 (EP_1_ agonist), ONO-AE1-259 (EP_2_ agonist), ONO-AE-248 (EP_3_ agonist), or ONO-AE1-329 (EP_4_ agonist) at 0.1 μM with or without ATP (100 μM) for 3 h. The levels of (**A**) IL-6, (**B**) IL-13, and (**C**) TNF-α in the reaction medium were measured by ELISA. Values are shown as mean ± SEM (*n* = 3). (**D**–**F**) The BMMCs were preincubated with a vehicle, ONO-8713 (EP_1_ antagonist), AH6809 (EP_2_ antagonist), ONO-AE3-240 (EP_3_ antagonist), and ONO-AE3-208 (EP_4_ antagonist) at 1 μM for 5 min and then stimulated with PGE_2_ (0.1 μM) and ATP (100 μM) for 3 h. The levels of (**D**) IL-6, (**E**) IL-13, and (**F**) TNF-α in the reaction medium were measured by ELISA. Values are shown as mean ± SEM (*n* = 3).

**Figure 9 cells-11-00616-f009:**
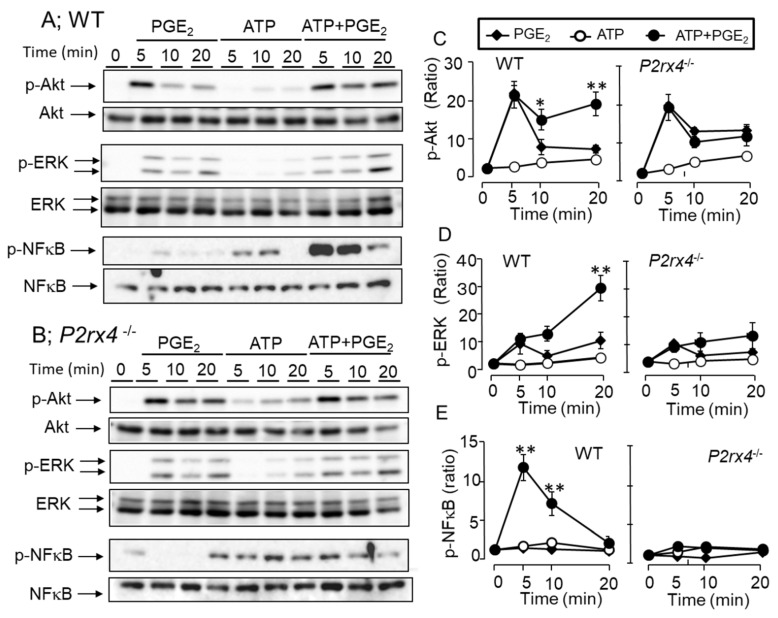
Effects of PGE_2_, ATP, and co-stimulation of PGE_2_ and ATP on ERK 1/2, Akt and NF-κB phosphorylation in the BMMCs. (**A**,**B**) The BMMCs prepared from WT (**A**) and *P2rx4*^−/−^ mice (**B**) were stimulated with PGE_2_ (0.1 μM), ATP (100 μM), and PGE_2_ and ATP for 5, 10, and 20 min. Cell lysates were subjected to Western blot analysis for phospho-Akt and total Akt (upper), phospho-ERK 1/2 and total-ERK 1/2 (middle), or phospho-NF-κB and total-NF-κB (lower). Blots are representative of three independent experiments. (**C**) The densitometry value of protein bands of phospho-Akt (**C**), phospho-ERK 1/2 (**D**), and phospho-NF-κB (**E**) are shown as relative intensities, with the results obtained with unstimulated cells (Time 0) designated as 1. Data are shown as mean ± SEM (*n* = 3). The effect of co-stimulation of PGE_2_ and ATP on ERK 1/2, Akt, and NF-κB phosphorylation is significantly different from that of PGE_2_ alone. * *p* < 0.05, ** *p* < 0.01.

**Figure 10 cells-11-00616-f010:**
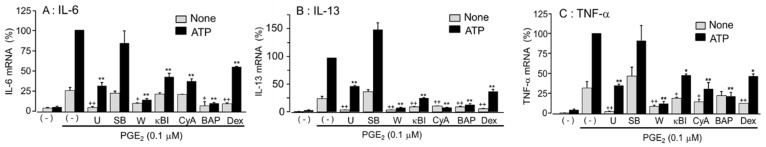
Effects of various inhibitors on the elevation of cytokine mRNA levels induced by the co-stimulation with ATP and PGE_2_. (**A**–**C**) The BMMCs were preincubated with a vehicle, MEK1/2 inhibitor U0126 (U, 5 μM), p38MAPK inhibitor SB203580 (SB, 10 μM), PI3K inhibitor wortmannin (W, 0.1 μM), NF-κB activation inhibitor (κBI, 1 μM), calcineurin inhibitor cyclosporine A (CyA, 5 μM), intracellular Ca^2+^ chelating agent BAPTA-AM (5 μM) and steroidal anti-inflammatory dexamethasone (Dex, 0.1 μM) for 10 min, followed by stimulation with PGE_2_ (0.1 μM) with or without ATP (100 μM) for 1 h. mRNA expression of (**A**) IL-6, (**B**) IL-13, and (**C**) TNF-α was examined by quantitative RT-PCR. Data were normalized with GAPDH mRNA levels and shown as the percentage of the response to PGE_2_ and ATP. Values are shown as mean ± SEM (*n* = 6). * *p* < 0.05, ** *p* < 0.01 vs. the response to ATP and PGE_2_ and ^+^
*p* < 0.05, ^++^
*p* < 0.01 vs. the response to PGE_2_.

## Data Availability

The data presented in this study are available in the main text.

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
