# Peer review of "Synergistic Cytokine Production by ATP and PGE2 via P2X4 and EP3 Receptors in Mouse Bone-Marrow-Derived Mast Cells"

_cells, 2022, doi:10.3390/cells11040616_

Round 1
Reviewer 1 Report
In this study, the authors demonstrated that activation of BMMC by ATP and PGE2 or Ag markedly increased the production and released of differences cytokines such as IL-6, TNF-a and IL-13. They showed that this effect was mediated by P2X4R and EP3R, respectively. Finally, they also investigated that intracellular signaling pathways involved in this MC activation pathway and fined involvement of NF-kB and ERK/PI3K / AKT phosphorylation.
My comments are:
- The authors shown in figures 1 and 2 that MC activated by co-stimulation of ATP and Ag or PGE2 increased mRNA cytokines expression after only 1hr and released of the cytokines after only 3hr. This is very rapid. The authors need to discuss these findings.
- In figure 3 the authors represent the kinetic of the effect of ATP or PGE2 alone or ATP and PGE2 together on the ability to induced cytokines release. What about the effect of ATP with Ag? Sinch it is known that Ag can induce cytokines release at time depended it interesting to examine the effect of ATP on time course of Ag-induced cytokine secretion as well.
- In figure 9, Can the authors provide a densitometry analysis of the blots 9A and 9B (including statistics) in order to better described the changes in the kinetics of pERK and pAKT between ATP or PGE2 along or together.
- In line 331 it described that " ATP had little effect on ERK and AKT phosphorylation at 2 min" and in the figure the times that are mentions are 5, 10 and 20 min. please correct or explained .
- The concentration of the Ag (anti DNP IgE) should be added to the m&M.
Reviewer 2 Report
The authors describe that ATP together with PGE2 results in a strong cytokine response. Thereby ATP only in combination with PGE2 induces a P2X4-dependent activation of PKB/Akt and ERK which are critically involved in the induced cytokine responses in mast cells. Though the results are of interest the mechanism behind this cytokine response is only poorly investigated and needs further work.
Figure 5, 6 and 7
Compared to Figure 1, 2, 3 and 4 the PGE2 concentration was changed to from 1µM to 0,1µM.…..Why?
Are there no effects of the antagonist when 1µM PGE2 was used? This reviewer thinks that it is mandatory for the authors to use only one concentration of PGE2 throughout the manuscript. Therefore it is required to repeat parts of the experiments so that only one concentration of PGE2 is used throughout the manuscript.
Figure 9
Again compared to Figures 1-4 0,1µM PGE2 was used. Why?
In the whole manuscript there is no evidence for p2x4 deficiency. This can easily be done by westernblotting or flow cytometry. In Figure 9 a WB for p2x4 is missing.
Furthermore is it essential to show whether the expression of EP3 receptors is similar in wt and p2x4-/- BMMCs.
The quantification of the pERK westernblotting experiments is wrong. The authors write in the legend that the PGE2 stimulation was set as 1 but in Figure 9E the PGE2 stimulation is around 0.8.
The data show that ATP alone does not induce its signaling pathways via P2XR4. Only when costimulated with ATP and PGE2, P2XR7 deficiency reduced the induced signaling pathways indicating that only co-stimulation induces a P2X4-dependent activation of Akt and ERK1/2. The authors only mention this fact on page 11, lane 426-428 but give no explanation for this finding. However, this is essential, when the authors claim that ATP via P2XR4 mediates an ERK and PI3K dependent production of cytokines in presence of PGE2. Given that this is an interesting and central point in this manuscript, this finding needs to be experimentally elucidated. Does PGE2 enhance the affinity of P2XR4 for ATP to facilitate the additional activation of ERK and AKT? Or does ATP initially act via P2XR7 and switches to P2XR4 when co-stimulated with PGE2? Or does PGE2 quickly upregulate the surface expression of P2X4 and thus sensitizes mast cells for ATP.
Figure 10
The authors claim the involvement of NFkB but in this manuscript the authors dont show which ligand either ATP and/ or PGE2 activates NFkB. This need to be investigated and can easily be done by westernblotting against phospho-p65!
In a recent paper the authors show that ATP and PGE2 alone induces a Ca2+ influx which is increased in response to co-stimulation with ATP and PGE2 (Yoshida et al., International Journal of Molecular Sciences). Therefore, it is interesting that the CsA inhibitor did not influence the cytokine production indicating that NFAT is not activated. However, the authors should use an alternative CsA inhibitor. If there is still no influence of CsA inhibition the authors should explain why?
Furthermore, if there is a Ca2+ release, it is required and of importance to investigate whether Ca2+ is necessary for the activation of Akt and ERK1/2 and the resulting cytokine production. This can be done by using Ca2+ chelating BAPTA-AM.
Jordan et al. 2021, Immunology, recently showed that the ATP-induced cytokine production in mast cells strongly depends on Ca2+/CsA and NFATc2. In contrast to this, in this manuscript the authors claim that there is an ATP/PGE2-induced Ca2+ response which did not induce a CsA-dependent cytokine response which excludes NFAT activation. How do the authors explain the differences between their manuscript and the work from Jordan et al. 2021. The authors should address these differences in detail in the discussion.
Round 2
Reviewer 2 Report
Figure 10.
The authors wrote as an answer:
Answer: Thank you for important advice. As mentioned above, we investigated the phosphorylation status of NF-kB and found that co-stimulation of ATP and PGE2 rapidly and significantly increased phosphorylation of NF-kB. This result was added Figure 9 A and B with quantitative densitometry analyses (E). The Figure 9B-C in the original version have been removed to avoid duplication and is shown in the Supplementary Figure S2 with the results of pNF-kB and P2X4R. In this regard, we have revised corresponding paragraphs in Methods (lines 88-91, 97-98, 100-101), Results (lines 320-330), Discussion (lines 448-456), and Figure legends (347-354).
The reviewers statement:
Thank for these blots. However, compared to wt mast cells in PGE2-stimulated P2X4Kos there is a loss of the p65 phosphorylation (now Figure 9B). How does the authors explain the loss of pNFkB phosphorylation in P2X4Kos?
Which pNFkB antibody was used?
Author Response
Thank you for important advice. As mentioned above, we investigated the phosphorylation status of NF-kB and found that co-stimulation of ATP and PGE2 rapidly and significantly increased phosphorylation of NF-kB. This result was added Figure 9 A and B with quantitative densitometry analyses (E). The Figure 9B-C in the original version have been removed to avoid duplication and is shown in the Supplementary Figure S2 with the results of pNF-kB and P2X4R. In this regard, we have revised corresponding paragraphs in Methods (lines 88-91, 97-98, 100-101), Results (lines 320-330), Discussion (lines 448-456), and Figure legends (347-354).